# Multiplexed coding by cerebellar Purkinje neurons

**Sungho Hong[1]\*, Mario Negrello[1,2], Marc Junker[3], Aleksandra Smilgin[3], Peter Thier[3], Erik De Schutter[1]**

[1]Computational Neuroscience Unit, Okinawa Institute of Science and Technology, Okinawa, Japan; [2]Department of Neuroscience, Erasmus Medical Center, Rotterdam, The Netherlands; [3]Department of Cognitive Neurology, Hertie Institute for Clinical Brain Research, University of Tübingen, Tübingen, Germany

**Abstract** Purkinje cells (PC), the sole output neurons of the cerebellar cortex, encode sensorimotor information, but how they do it remains a matter of debate. Here we show that PCs use a multiplexed spike code. Synchrony/spike time and firing rate encode different information in behaving monkeys during saccadic eye motion tasks. Using the local field potential (LFP) as a probe of local network activity, we found that infrequent pause spikes, which initiated or terminated intermittent pauses in simple spike trains, provide a temporally reliable signal for eye motion onset, with strong phase-coupling to the β/γ band LFP. Concurrently, regularly firing, non-pause spikes were weakly correlated with the LFP, but were crucial to linear encoding of eye movement kinematics by firing rate. Therefore, PC spike trains can simultaneously convey information necessary to achieve precision in both timing and continuous control of motion.

\*For correspondence: shhong@oist.jp

**Competing interests:** The authors declare that no competing interests exist.

## Introduction

Movements are often executed with high precision in timing and trajectory control. The cerebellum is heavily involved in online motor control and should process sensorimotor information with great accuracy. In particular, PCs, which deliver the final output from the cerebellar cortex, should use an appropriate coding strategy for this task, but the nature of their coding mechanism is actively debated (*De Zeeuw et al., 2011*; *Heck et al., 2013*). In one view, transmission of timing-sensitive information is performed by precisely timed PC spikes and their synchronized firing (*Ebner and Bloedel, 1981*; *Gauck and Jaeger, 2000*; *Shin and De Schutter, 2006*; *de Solages et al., 2008*; *De Zeeuw et al., 2011*; *Person and Raman, 2012*). In the other, PCs use *linear* firing-rate coding with weak PC-to-PC correlations to robustly control continuous movement kinematics, where high signal-to-noise ratio is achieved by averaging the rates of many PCs (*Shidara et al., 1993*; *Thier et al., 2000*; *Roitman et al., 2005*; *Medina and Lisberger, 2007*; *Catz et al., 2008*; *Herzfeld et al., 2015*).

In this study, we re-examined this controversy using a new approach to analyze PC spike trains. We classified PC spikes into specific spike categories, and correlated spike categories with the LFP, using the LFP as a proxy signal for local network activity. In particular, we focused on the role of long, infrequent interspike intervals (ISI), called pauses, which abruptly interrupt the rapid and very regular firing of PCs (*Schonewille et al., 2006*; *Shin and De Schutter, 2006*; *Shin et al., 2007*; *Yartsev et al., 2009*). Pauses in the PC spike train are a well-known phenomenon in many contexts, such as saccades (*Ohtsuka and Noda, 1995*; *Arnstein et al., 2015*; *Herzfeld et al., 2015*), and classical conditioning (*Rasmussen et al., 2008*), etc. Spikes that initiate and terminate pauses often synchronize sharply across nearby PCs (*Shin and De Schutter, 2006*). This suggests that pauses can be

**eLife digest** The cerebellum is a part of the brain that uses information from the senses to coordinate movement. Cells called Purkinje neurons in the cerebellum produce the final 'output' of its cortex. Therefore, Purkinje neurons have to communicate precise information about different aspects of the movement, such as its speed and timing. This information is likely to be represented by patterns of electrical activity within Purkinje neurons, but these patterns are still not fully understood.

Hong et al. recorded and analyzed electrical 'spikes', the output activity of Purkinje neurons, while monkeys made rapid eye movements. The recordings showed that occasional pauses in the otherwise regularly firing spikes of Purkinje neurons signaled the start of the eye movements. The pauses were accompanied by a sharp change in the local field potential, another electrical signal that comes from many neurons in the neighborhood. In the same cells, the rate of regularly firing spikes increased and decreased with the direction and speed of eye movements, following a simple relationship and independently of the local field potential.

Purkinje neurons therefore appear to use both the timing and the rate of their spiking activity to represent movement. This resolves conflicting reports in the literature claiming that either rates of spiking or their timing code essential information about movements: both are important. This way of representing information by combining more than one source is known as multiplexed coding.

Next, experiments recording electrical activity from many cells in the cerebellum at the same time are needed to find out how multiple Purkinje neurons can pause their spiking activity at the same time. Future experiments should also uncover how pauses in spiking and firing rates change with learning.

simultaneously involved in spike coding by individual PCs and with collective encoding in a local network.

Our approach was to examine the relationships among PC spikes, cerebellar LFPs, and eye motion, with a focus on how those relationships change, depending on the spike category. Using this, we demonstrate that PC spikes simultaneously contribute to precision, both in timing and control of motion by adaptive use of synchrony/spike time and rate coding scheme.

## Results

### Cerebellar LFP and single PC firing correlate to saccadic eye movements

From three rhesus (*Macaca mulatta*) monkeys (E, H, and N), we simultaneously recorded spikes of single PCs, the slow component of the LFP below low-$\gamma$ frequency (42 Hz), and eye positions during spontaneous and visually guided saccades (*Figure 1A*). Only recordings where both simple and complex spikes could be isolated were used. In this study, we focused exclusively on simple spikes, since complex spikes comprised a very small percentage of total spikes ($1.71 \pm 0.13\%$ with a rate of $0.81 \pm 0.07$ Hz, mean $\pm$ SEM) and their overall effect on firing statistics was not significant (*Figure 1—figure supplement 1H*, see also below).

We first estimated the sensitivity of neural signals to eye motion by computing their cross-correlation functions to eye velocity (EV), $CCF_{LFP-EV}$ and $CCF_{Spike-EV}$ for the LFP and simple spikes, respectively (*Figure 1B*). Two sharp differences between the LFP and spikes were noticed: First, $CCF_{LFP-EV}$ was significant in recordings from 49 cells ($p<0.01$, *t*-test, $n = 33, 15, 1$ from E, H, and N, respectively), but significant $CCF_{Spike-EV}$ was found only in a subset of 34 cells ($p<0.01$, *t*-test, $n = 21, 12, 1$ from E, H, and N, respectively). These 34 cells were used in the rest of the analysis. Second, when computed to account for the eye speed component for specific directions, the CCF varied less significantly with the angles of eye movements for the LFP than for spikes, in most cases (*Figure 1B,C*). We also computed the average LFP and firing rates for saccades with different directions and durations, and found that they shared the same properties, having similar waveforms as the CCFs (*Figure 1—figure supplement 1A–F*).

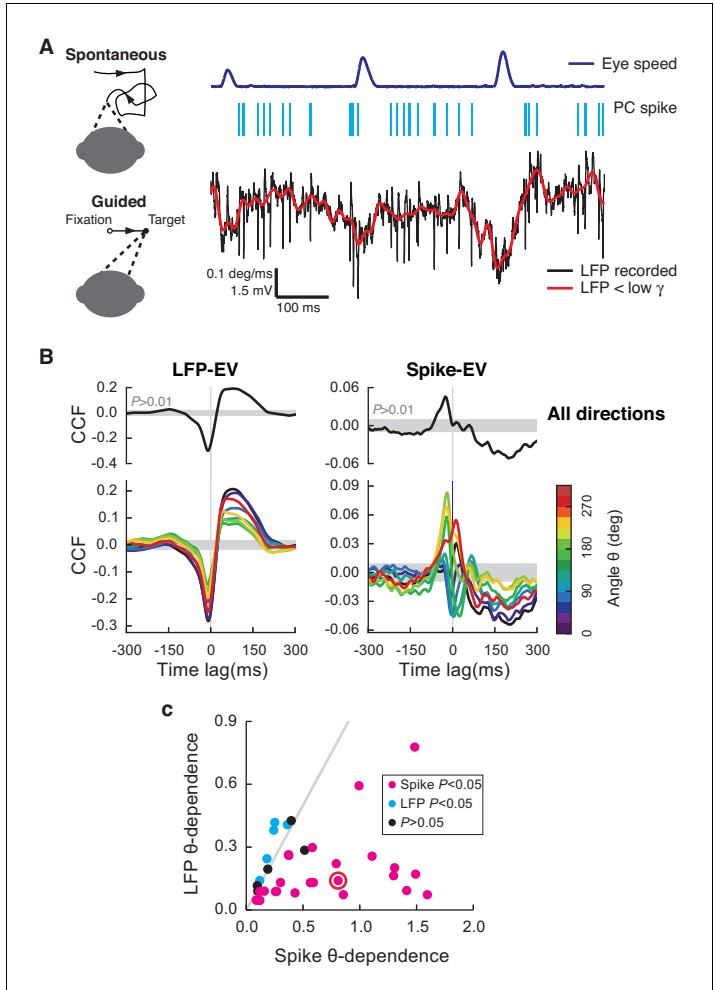

**Figure 1.** Cerebellar LFP and PC spikes correlate differently with saccadic eye movements. (**A**) Left: Schematics of eye motion tasks. Right: Simultaneously recorded eye speed, PC spike train, and LFP. Both the recorded (black) and filtered LFP below the low-γ frequency (red) are shown, but only the latter was analyzed. (**B**) $CCF_{LFP-EV}$ and $CCF_{Spike-EV}$ computed with EV (Top) and eye speed in the direction with the angle θ (Bottom). Shaded regions represent the 99% confidence interval. Data are normalized as described in the Materials and methods and plotted as mean ± SEM. (**C**) θ-dependent variability of $CCF_{LFP-EV}$ (x-axis) versus $CCF_{Spike-EV}$ (y-axis). $CCF_{Spike-EV}$ varied significantly more (p<0.05, *t*-test) than $CCF_{LFP-EV}$ in 74% (*n* = 25; magenta), and less in 15% (*n* = 5; cyan) of all recordings. No difference was found in the rest (*n* = 4; black). Error bars are omitted for clarity. The gray line represents equal variability. The red circle denotes data in A and B.

The following figure supplement is available for figure 1:

**Figure supplement 1.** Saccade angle and duration dependence of the onset-triggered average LFP, simple and complex spikes.

We identified eye saccade-related neurons based on $CCF_{LFP-EV}$ and $CCF_{Spike-EV}$; this is essentially identical to the classical method used to localize eye motion-sensitive cells (*Ohtsuka and Noda, 1992*). Differences in how LFP and PC spikes relate to eye movements were similar to previously reported differences between multiple PCs versus single units, where the averaged activity of multiple PCs is more robust, but much less direction-dependent (*Ohtsuka and Noda, 1995*; *Thier et al., 2000*). This supports the view that the LFP represents the average activity of many neurons with diverse angle-dependencies in the local network (*Buzsáki et al., 2012*) rather than reflecting the

activity of a single PC or a cluster of PCs with similar directional tuning. For this reason, the LFP and spikes apparently do not encode identical information about eye motion.

## PCs fire regularly most of the time, but occasionally pause

All recorded PCs fired rapidly (55 ± 18 Hz, mean ± SD) with high regularity. To quantify this regularity, we computed the ISI asymmetry index (AI) and coefficient of variation (CV$_2$) (*Shin et al., 2007*) for each data set, and compared them with those of control spike trains. In controls, spikes were randomly generated with the same instantaneous firing rates and refractory period (4 ms) as in the reference PC spike train. All of our spike trains contained a significantly higher number of regularly firing spikes (AI ≈ 0, or equivalently CV$_2$ ≈ 0) than the control (p<0.01, one-sided z-test; *Figure 2A*). ISI distributions had long tails that followed a power law, $P_{ISI}(x) \sim x^{-\alpha}$ ($\alpha$ = 3.78 ± 0.64; tail onset = 30.4 ± 13.4 ms; p<0.05, n = 32) (*Figure 2B*). Furthermore, the size of the ISI after each spike quickly diverged as the AI of the spike increased (*Figure 2C*), implying that deviation from regular firing tends to be associated with long ISIs.

Our observations are consistent with previous studies describing the PC spike train as typically composed of prolonged periods of fast and highly regular firing, occasionally interrupted by longer

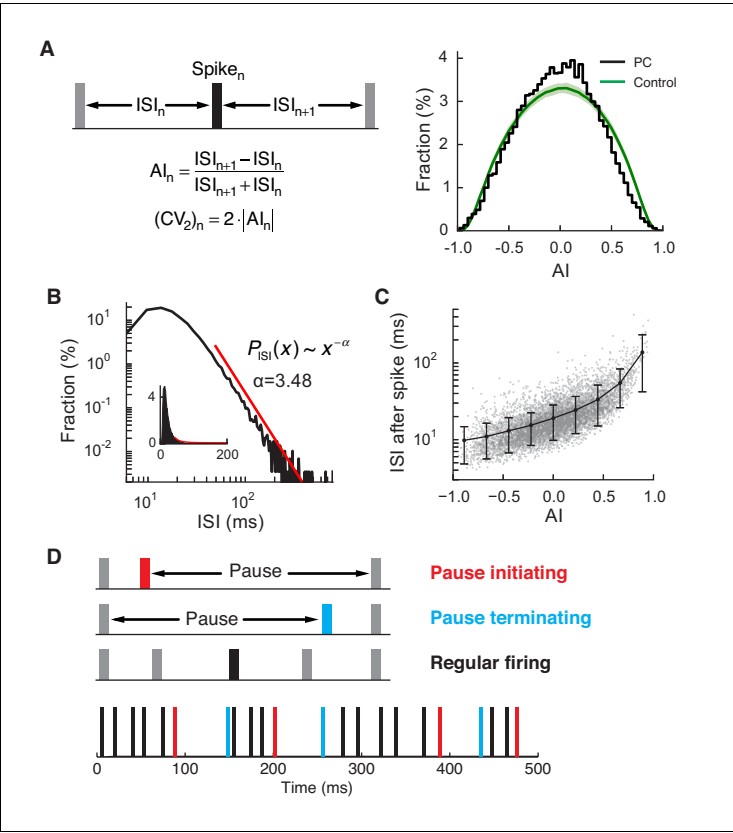

**Figure 2.** Regular and pause spikes in the PC spike train. (**A**) Left: The ISI asymmetry index (AI) measures local variability at each spike and is related to the local coefficient of variation (CV$_2$). Right: Distribution of AI for PC spikes (black) and rate-matching control spike train (green, mean ± SD). Significantly more PC spikes occurred around AI ≈ 0. (**B**) The ISI histogram with a fitted power-law tail (red). Inset: the same histogram and tail in linear scales. (**C**) ISI after each spike vs. AI. The black line represents mean ± SD in each bin (center = [-0.8, -0.6, . . ., 0.8], width = 0.2). (**D**) Top: Three types of spiking pattern classified by their AI and associated ISI length. Bottom: Examples of pause and regular spikes in an actual PC spike train. Data are the same as in *Figure 1A,B*.

The following figure supplement is available for figure 2:

**Figure supplement 1.** Complex and simple spike pauses.

ISIs, called pauses (*Schonewille et al., 2006*; *Shin and De Schutter, 2006*; *Shin et al., 2007*; *Yartsev et al., 2009*) (*Figure 2D*). Some previous studies discussed extremely long (>200 ms) pauses and related them to membrane bistability of the PC (*Loewenstein et al., 2005*; *Schonewille et al., 2006*). However, the majority of longer ISIs, or pauses, were much shorter (<100 ms), even when they were associated with a moderately large AI (*Figure 2C*). Note that these were therefore comparable or larger in duration to pauses triggered by complex spikes (*Latham and Paul, 1970*) (see also *Figure 1—figure supplement 1H*) and also those following simple spike bursts induced by optogenetic excitation (*Lee et al., 2015*).

Although complex spikes triggered pauses, their contribution to pauses in our study remained limited. Complex spikes had a larger average AI than simple spikes (complex: $0.36 \pm 0.12$, simple: $5.9 \times 10^{-4} \pm 7.7 \times 10^{-4}$), but their standard deviations were comparable (complex: $0.36 \pm 0.05$, simple: $0.31 \pm 0.04$). This implies that AI values were widely distributed in both simple and complex spikes. However, durations of complex-spike pauses did not increase as rapidly with AI as those of simple-spike pauses. In all cells, correlations between AI and $\log_{10}$ (ISI) were significantly smaller for complex spikes than simple spikes (complex: $0.24 \pm 0.15$, simple: $0.66 \pm 0.03$; $p<2.84 \times 10^{-12}$, Wilcoxon rank-sum test), which made complex-spikes pauses poorly predicted by AI (*Figure 2—figure supplement 1A*). Therefore, if we select ~10% of all ISIs as representative 'pauses', based on ISI size and the AI value of a preceding spike (see below), only $4.3 \pm 2.5\%$ of all pauses would be caused by complex spikes. For this reason, we did not include complex spikes and their associated pauses in our analysis beyond this point.

## Pause spikes couple strongly to the LFP and specifically to the β/γ band

Our first question was whether the relationship between spikes of individual PCs and the activity of the local network vary with specific spike categories, i.e., regular-firing or pause-related. To examine this, we first selected three subsets of simple spikes from each cell, representing regular, pause-initiating, and pause-terminating spikes, based on a criterion combining AI and ISI duration (see Materials and methods) that selected ~10% of all spikes in each category (*Figure 2—figure supplement 1B*). Then, we computed the spike-triggered average LFP ($STA_{LFP}$) (*Gray and Singer, 1989*; *Soteropoulos and Baker, 2006*) for each.

We found that the $STA_{LFP}$ of spikes that initiated or terminated pauses was sharply different from those of regular spikes: the pause-related $STA_{LFP}$ was characterized by strong and sharp peaks around the spike time, whereas the regular spike-related $STA_{LFP}$ showed only a weak modulation (*Figure 3A* left, B). For comparison, if computed with randomly selected, or all simple spikes, the $STA_{LFP}$ showed a level of peak correlation similar to that of regular spikes (*Figure 3A* right, B). This suggests that pause spikes are more strongly and more precisely coupled to network activity than regular spikes, but this fact goes unnoticed if the temporal structure of the spike train is ignored when computing the $STA_{LFP}$ (*Figure 3A* right), because pauses are relatively rare events. The larger amplitude of the $STA_{LFP}$ for pause-related spikes was readily observed when we varied the criterion for pause spike selection (*Figure 3—figure supplement 1*).

In a few exceptional PCs, the $STA_{LFP}$ of regular spikes had a modulation amplitude comparable to that of pause spikes (>70% in amplitude, $n = 5$; *Figure 3—figure supplement 2A*), whereas these cells did not show difference in correlation with eye movements (measured by max $|CCF_{Spike-EV}|$) from the rest ($p>0.16$, Wilcoxon rank-sum test). However, in most of these cases ($n = 4$), the $STA_{LFP}$ of regular spikes developed on a much slower time scale, mostly below the high β frequency (15 Hz). Therefore, when the $STA_{LFP}$ was recomputed with the β/γ band (15–42 Hz) LFP, $STA_{LFP}$ amplitudes were mostly retained for pause spikes, but significantly diminished for regular spikes, particularly if the original amplitude was large (*Figure 3—figure supplement 2B–D*). This was observed even when the LFP power spectrum did not have a distinct peak in the β/γ band, implying that it is a property of pause spikes rather than a product of LFP dynamics that generates the β/γ band oscillation.

The preferential coupling of pause spikes to the higher frequency LFP band, not the lower, led us to examine whether pause spikes are phase-locked to the β/γ LFP. This was true in the entire dataset, where pause spikes strongly preferred certain phases of the β/γ LFP while regular spikes showed only a weak dependence on the phase (*Figure 3C*). To estimate how reliably spikes fired at certain LFP phases, we computed the pairwise phase consistency (PPC), which is an average coincidence between any two spikes in the LFP phase space (see Materials and methods for more details) and

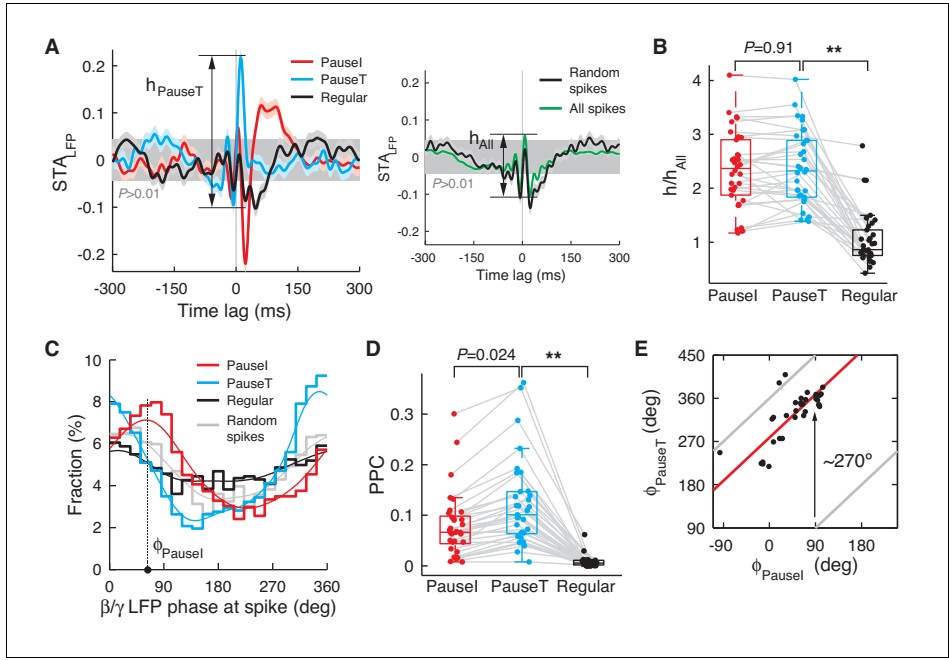

**Figure 3.** Pause and regular spikes correlate differently with the LFP. (**A**) Left: $STA_{LFP}$ of the pause-initiating (red), -terminating (cyan), and regular spikes (black). The grey region is the 99% confidence interval. $h$ is the peak-to-peak amplitude and shown for pause-terminating spikes. Right: $STA_{LFP}$ of randomly selected spikes (black) and all spikes (green). Data are mean ± SEM. (**B**) Relative STA amplitude, $h$, to that of all spikes ($h_{All}$) for each spike category. (**C**) Phase distribution of the β/γ LFP at the pause, regular, and randomly selected spikes (thick: histogram, thin: kernel-estimated density). φ denotes location of the peak, shown for the pause-initiating spike. (**D**) Pair phase consistency (PPC) for each spike category. (**E**) Peak phases for the pause-initiating (x-axis) and pause-terminating spikes (y-axis) in all data. Grey and red lines represent $\varphi_{PauseT} = \varphi_{PauseI}$ and $\varphi_{PauseT} = \varphi_{PauseI} + 278.3°$, respectively. In (**C**) and (**E**), **$p<10^{-5}$ (Wilcoxon rank-sum test). Data in (**A**), (**D**), and (**E**) are the same as in *Figure 1A,B*.

The following figure supplements are available for figure 3:

**Figure supplement 1.** $STA_{LFP}$ with different selection criteria for pause spikes.

**Figure supplement 2.** $STA_{LFP}$ in other data sets.

**Figure supplement 3.** Spike-LFP phase locking in other frequency bands.

**Figure supplement 4.** $STA_{LFP}$ depends on LFP spectral properties.

---

provides an unbiased measure of phase locking (*Vinck et al., 2010*). Again, we found a large difference between pause and regular spikes in their PPC (*Figure 3D*), specifically for the β/γ LFP (*Figure 3—figure supplement 3*). Notably, pause-initiating spikes preferentially occurred near the climbing phase, i.e., $<\varphi_{PauseI}> = 48.7° ± 45.7°$ where $\varphi_{PauseI}$ was the most preferred phase for each data set, and pause-terminating spikes were shifted by about 3/4 cycles on average ($<\varphi_{PauseT}> = <\varphi_{PauseI}> + 278.3° ± 35.3°$) (*Figure 3E*).

These results clearly show that pauses are not randomly occurring events. On the contrary, their initiating and terminating spikes are temporally locked to specific patterns of network activity, which generate significant and temporally precise fluctuations in the LFP.

# Pause spikes and correlated LFP components encode the timing of eye motion

If pause spikes in single PCs are related to specific patterns of network activity, what information do they jointly encode about eye motion? To answer this question, we first probed how the phase of the β/γ LFP evolves during each saccade in recordings for each cell. In many of them, the β/γ LFP was significantly coherent across all saccades, with a phase locking to saccade onsets (*Figure 4A,B*). We quantified this by computing the average cross-saccade phase coherence during a time window around saccade onset (from −100 ms to 150 ms), which we called LFP phase reliability. Significantly high LFP phase reliability was found in 80% of the data (p<0.01, $n = 28$, one sided *t*-test to time-shifted control data; $<R_{LFP}> = 0.20 \pm 0.02$, mean ± SEM).

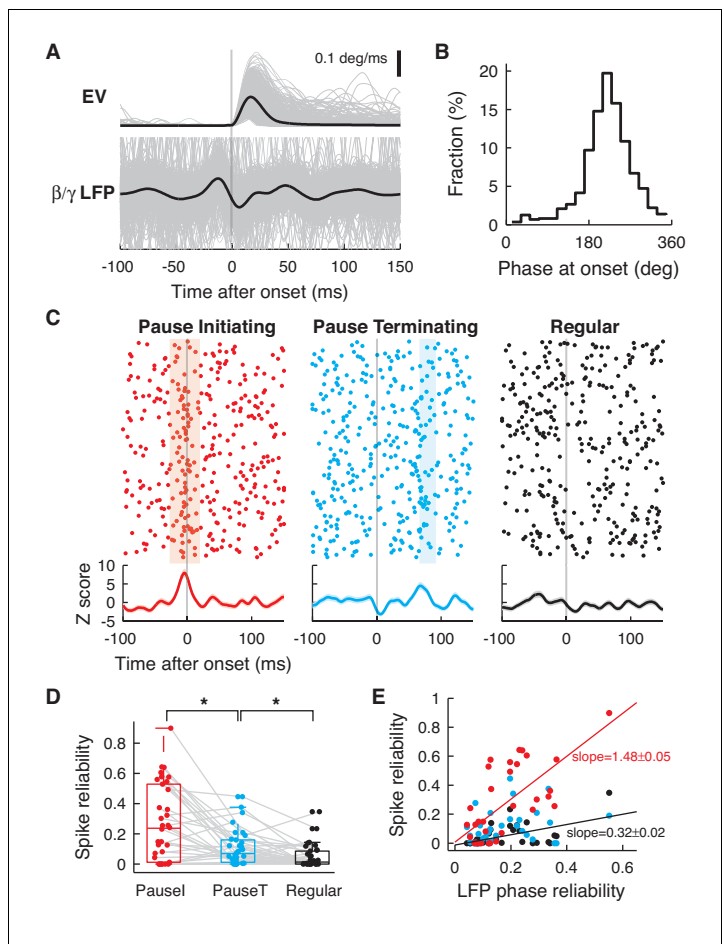

**Figure 4.** Pause spikes and β/γ LFP encode motion timing. (**A**) Eye speed (Top) and β/γ LFP (Bottom) aligned with saccade onset. Black lines represent an average over all saccades (grey, $n = 865$). (**B**) Phases of the β/γ LFP at saccade onset for the same data. (**C**) Top: Spike trains for pause-initiating, -terminating, and regular spikes aligned with onsets of randomly subselected saccades ($n = 289$). Light colored regions represent periods with significantly reliable firing ($jp(t)$<0.05, see Materials and methods). Bottom: Z-score for spike occurrence, smoothed by a gaussian kernel ($\sigma = 5$ ms). (**D**) Spike reliability of pause and regular spikes in all data. *p = 0.0229, 0.0193 (Wilcoxon rank-sum test). (**E**) LFP phase reliability versus spike reliability. The two were significantly correlated (red and black line) for pause-initiating and regular spikes (p = 9.3074 × 10⁻⁶, 0.0012; Fisher's z-test), but not for pause-terminating spikes (p = 0.0912), due to some recordings with low spike reliability but high LFP reliability. Data in (**A–C**) are the same as in *Figure 1A,B*.

The following figure supplement is available for figure 4:

**Figure supplement 1.** Peak firing of pause and regular spikes during saccades.

This predicted that pause spikes, which are phase-locked to the β/γ LFP, should reliably code eye movement timing. The most significant pattern that we observed was that pause-initiating spikes encode saccade onsets with a significant and sharp increase in average firing (*Figure 4C* and *Figure 4—figure supplement 1*). This could be caused either by a firing rate increase of pause spikes during some saccades, or by a sharp spike-time correlation across most saccade trials, i.e. reliable spiking. To ascertain which, we estimated how many spike coincidence events occurred beyond the prediction from the firing rate, from all possible pairings of spike trains and saccades, which we called the spike reliability: Briefly, for each pair of saccade trials, we counted spike coincidences ($|\Delta t| \leq 3$ ms) between two spike trains within a 50 ms-long moving time window, centered at $t$ ($-100$ ms $\leq t \leq 150$ ms, relative to saccade onset). Then we estimated the probability, $jp(t)$, that two random spike trains with the same firing rates could have the same or more spike coincidences, looking for significant ones ($jp(t) < 0.05$) (*Riehle et al., 1997*; *Denker et al., 2011*; *Ito et al., 2011*). The fraction of those significant coincidences from all cells, summed over all trials and time, became the spike reliability. Significantly coincident spikes were indeed found during significant peaks of firing probability (colored regions in *Figure 4C*); therefore, spike reliability was significantly higher for pause spikes (*Figure 4D*), further emphasizing their role in encoding the onset of eye motion. Finally, the LFP phase and spike reliability were much more steeply related to each other for pause-initiating spikes than for regular spikes (*Figure 4E*). This demonstrates that the fidelity of temporal coding by pauses in individual PCs crucially reflects the reliability of activity and coding of the local network.

## PC firing rate linearly encodes eye motion, mostly by regular spikes

So far we have focused on pause spikes, rare events in the spike train that provide timing information. On the other hand, eye velocity-spike correlation (and similarly saccade-triggered average rate) showed firing rate modulation by eye movement kinematics such as direction, duration, etc. (*Figure 1B–C* and *Figure 1—figure supplement 1A–F*). This suggests that a firing rate code may also be present in PC spike trains.

To answer this question, we constructed an inverse model that predicts firing rate from eye movements in each cell, based on the linear-nonlinear (LN) model framework (*Victor and Shapley, 1980*). LN models are widely used in sensory system studies, but have not been applied to PCs to date. First, the eye velocity profile is compared with a preferred pattern (motion feature) estimated from the linear cross-correlation (CCFs in *Figure 1B*) to generate a linear prediction $m$ of the firing rate, which goes through an additional nonlinear transformation to fit the actually observed firing rate (*Figure 5A,B*).

We found that the linear rate prediction $m$ followed the actual rate very closely ($R^2 = 0.92 \pm 0.07$, mean $\pm$ SD); therefore, the effect of the nonlinear transformation is small (*Figure 5C–E*). This made the linear rate prediction $m$ alone a good predictor of the time-course of average firing rate modulation during saccades and it was used exclusively in the rest of the analysis. Importantly, correlation with the firing rate remained high ($R^2 = 0.81 \pm 0.17$) when we computed the linear prediction with only regular spikes, which comprised about 20% of all spikes. Conversely, the firing rate of pause spikes (both pause-initiating and –terminating spikes) did not modulate as linearly or steeply as for regular spikes ($R^2 = 0.35 \pm 0.25$). This suggests that information encoded by the whole firing rate is very different from eye movement features related to pause spikes such as onsets (*Figure 4C*), but this information can be captured well by a subset of the full spike train, regular spikes.

This led us to further inquire into the similarity between rate coding by regular spikes and the full spike train. For this, we imposed different thresholds on $CV_2$ ($=2|AI|$) for selecting the regular spikes, computed the direction-dependent $CCF_{Spike-EV}$ of those selected spikes (*Figure 1B*), and estimated its similarity to that of the full spike train by computing their correlation coefficient $\rho$. We found that $\rho$ quickly grew as we increased the $CV_2$ threshold, whereas the fraction of regular spikes increased more slowly (*Figure 5—figure supplement 1*). For example, when we select ~47% of all spikes as regular spikes by imposing $CV_2 < 0.4$, their $CCF_{Spike-EV}$ is very similar ($\rho \approx 0.93$ in average) to that of the full spike train. Even at 20%, $CCF_{Spike-EV}$ of these regular spikes had $\rho \approx 0.85$ on average, similar to the linear coding property (*Figure 5C*). Note that $\rho$ for pause spikes behaved very differently; it was not significant for a reasonable fractions of pause spikes (*Figure 5—figure supplement 1* inset).

The combination of *Figure 5* and *Figure 5—figure supplement 1* demonstrates that regular spikes dominate rate coding. Regular spike and full spike trains respond to similar motion features

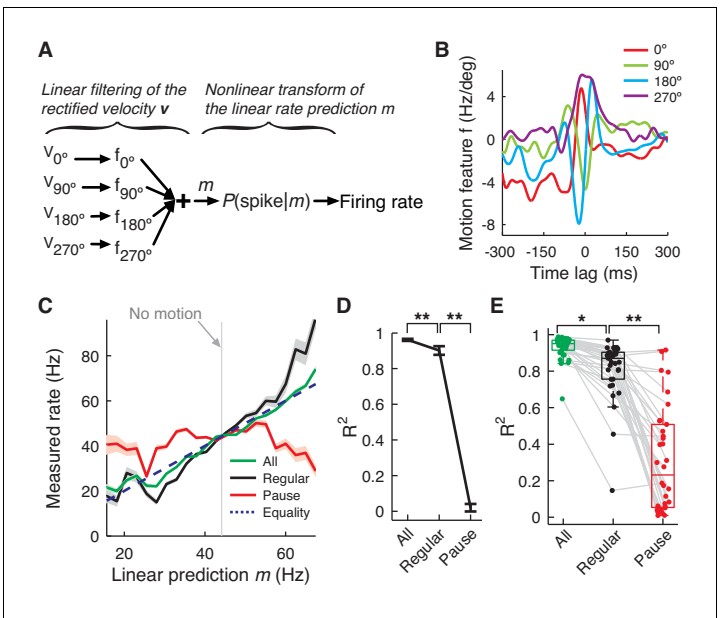

**Figure 5.** PC firing rate linearly encodes motion kinematics. (**A**) Schematics of the eye motion-to-rate inverse model. (**B**) Example motion feature. (**C**) Predicted vs. measured firing rate for all (green), regular (black), and pause (red) spikes. The rates of pause and regular spikes are rescaled to match those of all spikes to compensate for subsampling. The dotted line represents equality. (**D**) Goodness of fit $R^2$ for the linear prediction to actual rate. **p<$10^{-30}$ (Wilcoxon rank-sum test). (**E**) $R^2$ for all cells. Error bars are omitted and box plots are for means. *p<$10^{-6}$, **p<$10^{-11}$ (Wilcoxon rank-sum test). Error bars represent SEM. Data in B–D are the same as in *Figure 1A,B*.

The following figure supplement is available for figure 5:

**Figure supplement 1.** Spikes with high regularity have a spike-eye motion correlation that is very similar to that of the full spike train.

and also share the linear coding property. Our results resemble those of a previous report in which 'patterns' of regularly firing spikes dominate the rate responses of PCs to sensory stimuli (*Shin et al., 2007*). Nevertheless, pause spikes can make small contributions to rate coding. With all spikes, the LN model is empirically linear, but the rate coding of regular spikes predicted by the model is slightly less linear (*Figure 5D,E*). Pause spikes, which by definition represent fast rate changes, must compensate for this.

One regular spike is inexpensive from the standpoint of information content, because a small and very regular subset can already provide a good approximation of the full rate modulation (*Figure 5— figure supplement 1*). Furthermore, regular spikes are weakly coupled to network activity (*Figure 3*). This evidence strongly suggests linear rate coding as the primary role of regular spikes.

## Discussion

Examples in sensory and motor systems (*Riehle et al., 1997*; *Panzeri et al., 2010*; *Gire et al., 2013*) and theoretical analysis (*Ratté et al., 2013*) have shown that neurons can use multiplexed coding strategies, where each spike in a single spike train can differentially couple to local network activity and encode different information about sensory stimuli or behavior. This study presents the first evidence for multiplexed coding in cerebellar Purkinje cells. Specifically, spikes that initiate pauses are strongly coupled to the β/γ band of the LFP and are therefore probably synchronized among nearby PCs. These spikes form a temporally reliable signal to initiate saccadic eye motion. Conversely, regular spikes in the same spike trains are desynchronized among nearby PCs and form a rate code that predicts direction selective eye kinematics. Use of a multiplexed code resolves the perceived

contradiction between temporal and rate coding that has dominated recent discussions about PC spiking (*De Zeeuw et al., 2011*; *Heck et al., 2013*).

Pauses in the spike train are observed in many tonically firing neurons in various contexts. In striatal cholinergic interneurons, synchronized pauses after bursts encode a salient stimulus (*Aosaki et al., 1995*). Since pause-initiating and -terminating spikes can synchronize sharply (*Jaeger, 2003*; *Shin and De Schutter, 2006*), PCs can potentially operate by a similar coding mechanism. In fact, spike synchronization by PCs is a powerful mechanism to control their postsynaptic targets in the cerebellar nucleus (CN). With exceptionally fast GABAergic synapses (*Person and Raman, 2012*), CN neurons can reliably generate time-locked rebound spikes in response to synchronized inputs followed by simultaneous disinhibition, even at moderate levels of synchrony ($\leq$50%) (*Person and Raman, 2012*) and spike time jitter ($\leq$20 ms) (*Gauck and Jaeger, 2000*; *Sudhakar et al., 2015*). Furthermore, recent optogenetic experiments have shown that synchronous pauses induced either by direct/indirect inhibition or at the offset of direct excitation, can reliably trigger firing in CN neurons, and importantly, at movement onset (*Heiney et al., 2014*; *Lee et al., 2015*). Crucially, excitation-induced pauses were short (~35 ms), but effective, and even shorter pauses from direct inhibition caused similar effects (*Lee et al., 2015*).

Synchronized pause spikes (*Shin and De Schutter, 2006*) can not only represent timing information, but also can complement a rate code. Previous studies have analyzed PC firing collected over many trials and neurons, and found that a collective representation of time and motion emerges as a form of burst firing alone (*Thier et al., 2000*) or together with suppressed/paused firing of many PCs (*Catz et al., 2008*; *Arnstein et al., 2015*; *Herzfeld et al., 2015*). Our results suggest that pause spikes can enhance temporal fidelity of such population-level representations since their temporal consistency across different PCs (e.g. *Figure 4—figure supplement 1B*) can offer a reliable representation, despite large heterogeneity in rate coding schemes of different PCs. For example, *Herzfeld et al. (2015)* found that population PC coding of saccade direction depends critically on the timing of pause onset, which varies only up to ~10 ms. Here we showed that pause spikes of individual PCs can indeed fire with a reliability of a few milliseconds. This can be particularly important at the single trial/saccade level, but averaging over multiple trials and cells may not be a suitable approach to probe it. Instead, we evaluated the correlation between simultaneously recorded PC spikes and LFP, and also trial-to-trial correlation (reliability) of those signals, for each cell. Our results show that synchronized pause spikes constitute the most significant population signal in PCs despite their sparse appearance in individual spike trains, suggesting that they can be a specific signaling mechanism in the PC-CN part of the motor pathway.

The spike-LFP relationship indicates that pause spikes are generated by the local network. While firings by presynaptic afferents, granule cells, Golgi cells, as well as by the postsynaptic targets, are clearly related to the LFP (*Soteropoulos and Baker, 2006*; *Dugué et al., 2009*; *Ros et al., 2009*), PCs only occasionally or weakly modulate their simple spike firing with the LFP (*Courtemanche et al., 2002*; *Ros et al., 2009*), except for the very high frequency component (~200 Hz) (*de Solages et al., 2008*). Here we found that the β/γ LFP robustly and preferentially couples to pause spikes and that both can reliably encode time information in a correlated way. Considering that cerebellar LFP is coherent with neocortical LFP (*Courtemanche and Lamarre, 2005*; *Soteropoulos and Baker, 2006*; *Ros et al., 2009*) and also plays a critical role in maintaining LFP coherence between the sensory and motor cortex, particularly in the low γ band (*Popa et al., 2013*), pause spikes may also have a special relationship with the LFP in the cerebral cortex.

We did not attempt to resolve the origin of the time encoding LFP signal due to limitations of the experimental setup. However, there are multiple possible primary sources. One is the dense activation of mossy fibers and granule cells that accompanies the significant LFP in the granular layer (*Morissette and Bower, 1996*; *Roggeri et al., 2008*; *Diwakar et al., 2011*). Because our electrodes are probably too far from the granular layer to detect the signal directly, it is likely that that localized massive activity propagates via ascending and parallel fibers to activate many interneurons and ultimately PCs. In particular, molecular layer interneurons (MLI) could provide significant feedforward inhibition (*Mittmann et al., 2005*), causing PCs to pause (*Mittmann and Häusser, 2007*). If simultaneous activation of local MLIs (and/or their synaptic inputs to the local population of PCs) contributes to the LFP, this would explain our observed correlation of the fast LFP signal and pauses in PCs. However, the time-encoding signal components we observed seem to be highly localized in the cerebellar cortex. In LFPs recorded ~1 mm horizontally from the spike electrode, $STA_{LFP}$ amplitude,

particularly of pause spikes was greatly diminished to absent (*Figure 3—figure supplement 4*). This suggests that the pause-related β/γ LFP signal originates from localized sources and decays quickly with distance, spreading at most a few hundred microns. This is consistent with the fact that PCs are rarely synchronized unless they are very close to each other (<100 µm) (*Ebner and Bloedel, 1981*; *Jaeger, 2003*; *Shin and De Schutter, 2006*).

We also found that the effect of complex spikes was minimal since pauses triggered by complex spikes (*Latham and Paul, 1970*) had a distinct distribution compared to simple-spike pauses (*Figure 2—figure supplement 1A*). The overall impact of complex spikes was negligible, most probably because none of our tasks involved sensorimotor learning where complex spikes are crucial (*Catz et al., 2005*; *Medina and Lisberger, 2008*), as they tend to occur after saccades, triggered by significant saccade errors (*Herzfeld et al., 2015*).

Many studies have reported significant correlated spiking in similar settings. In the motor and visual cortex, synchronized spikes have larger $STA_{LFP}$ and better phase locking to the motion-related LFP β oscillation (*Denker et al., 2011*; *Ito et al., 2011*). In another part of the cerebellum, Medina and Lisberger found that firing rate variability and cross-correlation of PC firing both peaked at the onset of smooth-pursuit eye motion (*Medina and Lisberger, 2007*). Our findings are consistent with their observations since pause spikes, which fire reliably with respect to motion onset, have higher ISI variability, and significant coupling to the LFP implies correlated firing. Medina and Lisberger also suggested that such correlated spiking is due to common input to PCs (*Medina and Lisberger, 2007*), which can significantly contribute to the LFP signal (*Denker et al., 2011*) and trigger synchronized pauses (*Jaeger, 2003*).

On the other hand, PCs also use linear rate coding of eye movements and here non-pause regular spikes are predominant. Linear coding of motion kinematics by the PC firing rate has been repeatedly observed (*Shidara et al., 1993*; *Roitman et al., 2005*; *Medina and Lisberger, 2007*; *Herzfeld et al., 2015*). If correlations with other PCs are weak, as is the case for regular spikes, rate coding by individual PCs can precisely control continuous movements since CN neurons receive inputs from many PCs and can average away noise in individual inputs.

Because different stages of a movement may demand that some aspects of the motion be controlled more precisely than others, it is useful for PCs to use different spike codes. Our study demonstrates that PCs can use both temporal- and rate-coding schemes to multiplex their population output with different types of information (*De Schutter and Steuber, 2009*; *Ratté et al., 2013*), so that precision in motion timing and continuous control can be managed adaptably. We conclude that multiplexed coding is used for sensorimotor coordination in the cerebellar cortex.

## Materials and methods

### Electrophysiological recording and behavioral procedure

All animal experiments were approved by the local animal care committee (Protocol number: Regierungpräsidium Tübingen N1/08 and N6/13), conducted in accordance with German law and the National Institutes of Health's *Guide for the Care and Use of Laboratory Animals* and carefully monitored by the veterinary administration (Regierungspräsidium and Landratsamt Tübingen). Three adult male rhesus (*Macaca mulatta*) monkeys (E, H, and N; 10, 11, and 15 years old, respectively) were subjects in this study. They were implanted with a magnetic scleral search coil to record the eye position (*Judge et al., 1980*), a titanium head post to painlessly immobilize the head during experiments, and a circular titanium recording chamber located over the midline of the cerebellum to allow electrophysiological recordings (*Thier and Erickson, 1992*). Position and orientation of the implants were carefully planned using pre-surgical MRI and confirmed using postsurgical MRI that helped to direct electrodes to the oculomotor vermis. All surgical procedures were conducted using aseptic techniques under the full anesthesia consisting of isofluorane supplemented with remifentanil (1–2 µg/kg/min). All relevant physiological parameters such as body temperature, heart rate, blood pressure, $pO_2$, and $pCO_2$ were monitored. Postoperatively, buprenorphine was given until no sign of pain was evident. Animals were allowed to fully recover before starting the experiments.

Animals made saccades either without (*n* = 16) or with visual cues (*n* = 18), but we analyzed all data combined. In the spontaneous saccade paradigm, animals made eye movements freely in the absence of any given visual stimulus. In the visually guided task, animals first focused on a fixation

spot at the center of the monitor for periods of time that varied randomly between 1000 and 1500 ms. As soon as the fixation spot disappeared, a target for the primary saccade appeared randomly in one of eight possible locations in the periphery with the angle θ = 0°, 45°,... , 315° at a constant distance from the center, varying between 2.5° and 20° in separate blocks.

## Processing and selection of recording data

Extracellular potentials were initially high- and low-pass filtered online and recorded separately. The band pass-filtered (300 Hz–3 kHz) channel was used to identify single-unit activity. Single-PC units were distinguished by the presence of simple and complex spikes via online sorting, but all spikes were resorted offline and semi-automatically via a neural network trained on manually selected spike waveform prototypes as in (*Yartsev et al., 2009*).

The LFP from the low-pass filtered (<150 Hz) channel went through a series of additional filtering steps to remove influences from spike waveforms. First, an online notch filter at 50 Hz was applied to reduce line noise. An additional offline low-pass filter at 42 Hz was applied. Then, the LFP was resampled at 90 Hz and oversampled back to 1 kHz to minimize the effect of spike waveforms.

Eye velocity was computed from recorded eye position using a Savitzky-Golay filter of the fifth order with a 25 ms time window. Saccade onsets were detected from eye speed using a custom adaptive detection algorithm described in Appendix 1.

We first selected data based on recording quality both in neural and eye motion recordings and then based on statistical significance of $CCF_{EV\text{-}LFP}$ and $CCF_{EV\text{-}Spike}$ (see below).

## Calculation of cross-correlation function and spike triggered averages

When the signal for the quantity A and B are $x(t)$ and $y(t)$, respectively, the cross-correlation function $CCF_{AB}$ is computed by

$$CCF_{AB}(t) = \frac{1}{Z_{AB}}\sum_{s=0}^{L}x(s+t)y(s) - \frac{1}{M_{\text{shift}}}\sum_{m=1}^{M_{\text{shift}}}\left(\frac{1}{Z_{AB}}\sum_{s=0}^{L}x(s+t)y_{mT}(s)\right),$$

where $L$ is the signal length. $y_{mT}(t)$ is $y(t)$ shifted by $m \cdot T$, $y_{mT}(t) = y(t + m \cdot T)$. The second term is a shift-correction that estimates the baseline and average of possible uncontrolled correlation. We used $T = 1$ s and $M_{\text{shift}} \sim 200$ depending on $L$ and whether the variance of the shift corrections stabilized. Assuming this variance corresponds to the standard error of the CCF both for the original and shifted data, we computed the $t$-score for the one-sided test for whether the CCF becomes significant (p<0.01), particularly from $t = -100$ ms to $t = 100$ ms.

For the normalization $Z_{AB}$, we used two different schemes depending on whether A and B were both continuous (eye velocity and LFP) or one of them was a spike train. In the former, the 'CCF-like' convention, $Z_{AB} = (\text{Var}[x] \cdot \text{Var}[y])^{1/2} (L-|t|)$, was used. In the latter, we used the 'STA-like' normalization $Z_{AB} = \text{Var}[x]^{1/2} \cdot N_{\text{spike}}$ where $N_{\text{spike}}$ was the number of spikes. Therefore, $STA_{LFP}(t) = CCF_{LFP\text{-}Spike}(t)$ where $y(s)$ was a spike train with 1 ms wide time bins, up to a constant factor.

## Saccade-angle dependence of the LFP and PC spikes

We estimated dependence of the LFP and spikes on saccade angle θ by computing their CCF(θ) with the component of the eye velocity vector **v**, $v_\theta = (\mathbf{v} \cdot \mathbf{e}_\theta)_+$ where $\mathbf{e}_\theta = [\cos \theta, \sin \theta]$ (θ = 0°, 45°, ..., 315°) and $()_+$ represents rectification. Then, we computed the matrix **CCF** = [CCF(0°); CCF(45°); ...] and the noise-to-signal ratio, $\text{NSR} = \text{Tr Cov}[\mathbf{CCF}]/\|\text{Mean}[\mathbf{CCF}]\|^2$, which was used as an estimate of θ-dependence in CCF(θ). About 200 control CCF(θ) were made from time-shifted data (see above) and their variance was used in the $t$-test for differences in NSR between the LFP and spikes.

## Estimation of spike train statistics and selection of pause/regular spikes

The ISI-asymmetry index (AI) was computed as in *Figure 2A*. For each data set, we generated 200 rate-matching artificial spike trains in a similar way to *Shin et al. (2007)*. We first computed ISIs, $\{ISI_k\}$ ($k = 1,...N_{\text{spike}}\text{-}1$), from experimental data. At each $k$, we computed the local firing rate from the nearest five ISIs as $r_k = 5/(ISI_{k-2} + ISI_{k-1} + ... ISI_{k+2})$. The $k$-th artificial ISI, $T_k$, was drawn from the gamma distribution $P(T_k) \sim r_k^2 T_k \exp(-r_k T_k)$ with a refractory period 4 ms imposed. From these, we computed the mean histogram and its variance. The tail shape of the ISI distribution was tested using the *powerlaw* Python package (http://pypi.python.org/pypi/powerlaw), which provided a

fitting algorithm to the power-law tail and statistical tests to compare the result with the alternative hypothesis of an exponential tail.

For pause-initiating spikes, we first selected 15% of spikes having the largest AI values. Then, we removed 25% of spikes with the shortest pause ISI. Pause-terminating spikes were selected similarly, but with the smallest AI. Minimal pause duration varied with the average firing rate of the cell, but was in general ~20% larger than the ISI corresponding to the mean firing rate. Regular spikes were selected based only on their $CV_2$, and their number was matched to those in comparison groups.

### LFP phases in the β/γ band and their cross-saccade reliability

The β/γ LFP was obtained by band-pass filtering the LFP between 15–42 Hz. Phases were extracted by the Hilbert transformation (MATLAB function *hilbert*). From the phase and spike times, we computed pairwise phase consistency (PPC) in the following way (*Vinck et al., 2010*). When the phase of the β/γ LFP at spike times is $\{\varphi_n\}$ ($n = 1, \ldots, N_{\text{spike}}$), PPC is given by

$$\text{PPC} = \frac{2}{N_{\text{spike}}(N_{\text{spike}} - 1)} \sum_{m=1}^{N_{\text{spike}}-1} \sum_{n=m+1}^{N_{\text{spike}}} \exp(i(\phi_m - \phi_n))$$

LFP phase reliability $R_{\text{LFP}}$ is measured by phase coherence of the β/γ LFP across saccades. When $\varphi_i(t)$ is the LFP phase at $t$, time relative to saccade onset ($t = 0$ at the onset), for the $i$-th of $N$ saccades, $R_{\text{LFP}}$ is given by

$$R_{\text{LFP}} = \frac{1}{T_e - T_b} \int_{T_b}^{T_e} Z(t)dt, \quad Z(t) = \left| \frac{1}{N} \sum_{i=1}^{K} \exp(i\phi_i(t)) \right|,$$

where we used $T_b = -100$ ms and $T_e = 150$ ms. We also computed $R_{\text{LFP}}$ of time-shifted data in the same way as CCF and their mean and variance were used for the one-sided $t$-test.

### Cross-saccade spike reliability

Spike reliability $R_{\text{spike}}$ was evaluated using the fraction of significant cross-saccade synchronization events, following *Denker et al. (2011)*. We first collected spike trains during the same time window as for $R_{\text{LFP}}$. Then, within a 50 ms-wide moving window centered at $t$ in each saccade period ($t = 0$ at the onset), we counted the number of spike coincidences up to $\pm 3$ ms time difference, $n_{\text{emp}}(t)$. In the same moving window, rate prediction of the coincidence probability and expected number of coincidences $n_{\text{exp}}(t)$ were computed from firing rates based on summing all possible spike train/saccade pairings. With a null hypothesis of Poisson statistics, the probability of $n_{\text{emp}}(t)$ is

$$jp(t) = P\big(n_{\text{emp}}(t)|n_{\text{exp}}(t)\big) = \sum_{r=0}^{n_{\text{emp}}(t)} \frac{n_{\text{emp}}(t)^r}{r!} \exp\big(-n_{\text{exp}}(t)\big).$$

The criterion for significant synchronization was $jp(t) < 0.05$ and if this was satisfied, all coincidence events in the window were regarded as spike synchronization from 'unit events' (*Riehle et al., 1997*; *Denker et al., 2011*; *Ito et al., 2011*). Then, $R_{\text{spike}}$ is a fraction of synchronization events versus all coincidences,

$$R_{\text{spike}} = \frac{\sum\limits_{jp(t)<0.05} n_{\text{emp}}(t)}{\sum\limits_{t} n_{\text{emp}}(t)}.$$

### Eye motion-to-rate inverse model

Our inverse model consists of two parts: a motion feature **f** acts as a receptive field that linearly transforms eye velocity history to the linear rate prediction, *m*. Then, a nonlinear function $P(\text{spike}|m)$ gives the spiking probability. **f** was estimated from the spike-triggered average of the rectified eye velocity in four directions (θ = 0°, 90°, etc.) from −300 ms to 300 ms around the spike time. We computed the rate prediction, *m*, by linear filtering the rectified eye velocity by **f**, and estimated $P(\text{spike}|m)$ by comparing *m* with actual firing rate. See Appendix 2 for more mathematical details.

## Statistical analysis

When z- and t-tests were used, we checked normality of sample distribution using D'Agostino's $K^2$ test (significance level = 0.05). In some cases that failed the normality test, we estimated a p-value by computing an empirical upper bound of the type I error rate by using control data sets. We increased 400~2000 control data sets, depending on computing time, evaluated the quantity of interest, and made one-sided comparisons with an estimate from original data. The p-value was estimated to be an error rate of the comparisons.

For each statistical test, we computed statistical power by estimating an upper bound of the type II error rate from the resampled data sets: we generated 400~2000 randomly resampled data sets with replacement, performed the same statistical test on them, and counted how many passed the test at a given significance level, which gave our estimate of the power. We regarded the original test result as significant only when the power is sufficiently high (>0.8),

All analyses were done with MATLAB 2012a (Mathworks, VA) and custom scripts in Python 2.7, which will be available on our homepage (http://groups.oist.jp/cnu).

## Acknowledgements

We thank Sonja Grün, Chris De Zeeuw, Steve Prescott, Javier Medina, and Fahad Sultan for helpful discussions. We thank OIST Graduate University for its generous support of the Computational Neuroscience Unit. This work was also supported by grants from the Deutsche Forschungsgemeinschaft (Research Unit FOR 1847/1, project A3–TH 425/13-1, P Thier), the European Union (Marie Curie Training Network PITN-GA-2009-238214-C7, P Thier and A Smilgin) and the German Federal Ministry of Education and Research (Bernstein Center for Computational Neuroscience 01GQ1002A, project C3, P Thier, M Juncker and A Smilgin).

## Additional information

### Funding

| Funder | Grant reference number | Author |
| --- | --- | --- |
| Okinawa Institute of Science and Technology Graduate University | | Sungho Hong Mario Negrello Erik De Schutter |
| Deutsche Forschungsgemeinschaft | Research Unit FOR 1847/1, Project A3 (TH 425/13-1) | Peter Thier |
| Bundesministerium für Bildung und Forschung | Joint project Bernstein Center for Computational Neuroscience, 01GQ1002A, Project C3 | Marc Junker Aleksandra Smilgin Peter Thier |
| European Commission | PITN-GA-2009-238214-C7 | Aleksandra Smilgin Peter Thier |

The funders had no role in study design, data collection and interpretation, or the decision to submit the work for publication.

### Author contributions

SH, EDS, Conception and design, Analysis and interpretation of data, Drafting or revising the article; MN, Analysis and interpretation of data, Drafting or revising the article; MJ, AS, Acquisition of data, Drafting or revising the article; PT, Acquisition of data, Analysis and interpretation of data, Drafting or revising the article

### Author ORCIDs

Sungho Hong, http://orcid.org/0000-0002-6905-7932

### Ethics

Animal experimentation: All animal experiments were approved by the local animal care committee (Protocol number: Regierungpräsidium Tübingen N1/08 and N6/13), conducted in accordance with

German law and the National Institutes of Health's Guide for the Care and Use of Laboratory Animals and carefully monitored by the veterinary administration (Regierungsprääsidium and Landratsamt Tübingen).

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

# Appendix 1: Adaptive detection algorithm for saccade onsets

We first imposed a threshold $v_{th} = 10^{\mu+2.576\,\sigma}$ on the eye speed $v$, where $\mu$ and $\sigma^2$ are the mean and variance of $\log_{10}v$. This is based on our observation that the statistics of $\log v$ is approximately normal with jumps; therefore, $\log_{10}v_{th}$ corresponds to the threshold for outliers of $p<0.01$. Then, we detected the threshold breaking points, upward and downward. If the interval between the neighboring downward and upward breaking point was smaller than 75 ms, we removed those points altogether, which made all the inter-saccade intervals greater than 75 ms.

From these selected saccades, we detected candidate onsets. The eye speed, $v$, was high-pass filtered above 15 Hz and we looked for a negative peak in the filtered speed, $u$, within a 15 ms time window before the upward threshold breaking. This point, $t_{onset}$ marked the transition point to rapid eye motion and became the onset.

A small fraction of detected saccades had drifting eye motion before $t_{onset}$. We found the maximal eye speed between 12 ms and 2 ms before $t_{onset}$, which we call $v_{max}$, and excluded eye motions where $v_{max}$ exceeded an empirical threshold given by $v_{max,th} = \text{Median}[v_{max}] +2\Delta v_{max}$ where $\Delta v_{max} = |\text{Median}[v_{max}]\text{-Min}[v_{max}]|$. This criterion worked well empirically and removed $3.87 \pm 3.80\%$ of all detected saccades, which were ill-formed with slow drifting.

## Appendix 2: Construction of the inverse model and comparison with linear coding

### Inverse model

Our inverse model is based on the linear-nonlinear model scheme (*Victor and Shapley, 1980*) composed of a stimulus feature **f**, which acts as a linear filter on the incoming stimulus, **v,** to produce the 'motion variable', $m(t)$, and a nonlinear function $P(\text{spike}|m)$ that gives the probability to spike given $m(t)$, at each time bin (width = 1 ms) (*Figure 5C*).

Given eye velocity $\mathbf{u}(t) = [u_x(t), u_y(t)]$, we computed the rectified eye velocity in four directions, $\mathbf{v}(t) = [v_{0°}(t); v_{90°}(t); v_{180°}(t); v_{270°}(t)]$, where $v_\theta(t) = (\mathbf{u} \cdot \mathbf{e}_\theta)_+$, $\mathbf{e}_\theta = [\cos \theta, \sin \theta]$, and $()_+$ represents rectification. Then, the motion variable $m(t)$ is

$$m(t) = \int_{-\infty}^{\infty} d\tau \mathbf{v}(t - \tau) \cdot \mathbf{f}(\tau) \approx \int_{-T}^{T} d\tau \mathbf{v}(t - \tau) \cdot \mathbf{f}(\tau),$$

where $T$ is a time window size and we used $T$ = 300 ms.

### Estimation of the motion feature

First, we estimated the motion feature **f** by STA of the rectified eye velocity, **v**, i.e. STA$_\mathbf{v}$, equivalent to CCG$_\mathbf{v\text{-}Spike}$. This is similar to the linear regression approach for smooth eye movements (*Shidara et al., 1993*; *Roitman et al., 2005*; *Medina and Lisberger, 2007*; *Herzfeld et al., 2015*), whereas here our regression variables are the history of eye velocity for a extended period of time, not a small number of the motion variables, such as eye position, speed, acceleration, etc. In our case, eye position and speed or acceleration components correspond to the motion feature, **f**, being a broad integration kernel, a delta function-like single peak function or a biphasic differentiation kernel, respectively.

In practice, the spike train was converted to the firing rate series $r(t)$ by convolution with a Gaussian kernel with $\sigma$ = 10 ms, and the STA was computed with the weight of the smoothed firing rate. Then, **f** was estimated by the standard least square,

$$\mathbf{f}(\tau) = \int d\tau' dt \mathbf{W}_\alpha(\tau, \tau') \mathbf{v}(t - \tau') r(t),$$

where $\mathbf{W}_\alpha$ is the Ridge-regularized whitening matrix that compensates the autocorrelation of **v**,

$$\mathbf{W}_\alpha(\tau, \tau') = [\mathbf{C}(\tau, \tau') + \alpha \mathbf{1} \delta(\tau - \tau')]^{-1},$$

$$\mathbf{C}(\tau, \tau') = \int dt \mathbf{v}^T(t - \tau) \mathbf{v}(t - \tau').$$

For $\alpha$, we used the smallest eigenvalue of C among those that together captured 90% of the total variance, i.e., $\alpha = \lambda_n$ where $\sum_{i=1}^{n} \lambda_i / \text{Tr}\mathbf{C} \approx 0.9$ ($\lambda_i > \lambda_j$ for $i<j$). This choice gave us empirically good regularization, meaning that **f** was decorrelated, but still relatively free of artifacts (*Figure 5B*).

## Estimation of the nonlinearity

After obtaining the feature, **f**, we computed the linear rate prediction, $m(t)$, by filtering the rectified velocity **v** with **f**, and compared it with the rate $r(t)$ to get $P(\text{spike}|m)$. First, we computed $P(m|\text{spike})$ and $P(m)$ via kernel density estimation, where the former was weighted by the firing rate. Then, by Bayes' theorem,

$$P(\text{spike} \mid m) = \frac{P(m \mid \text{spike})P(\text{spike})}{P(m)}.$$

We repeated this procedure 500 times with randomly chosen bootstrap samples of $r(t)$ and $m(t)$ to get the bootstrap mean and SD of $P(\text{spike}|m)$ (**Figure 5C**).

In practice, finite size sampling noise in $P(\text{spike}|m)$ increases with $P(m)$ and thus $P(m|\text{spike})$ becomes small. To prevent this, we only used a range of the linear prediction variable, $[m_1, m_2]$, where $P(m|\text{spike}) > p_{\text{th}}$ for a threshold $p_{\text{th}}$. The threshold was lowered until the range contains as much as 99% of all samples of $m$ at spike times, i.e. $\int_{m_1}^{m_2} dm P(m \mid \text{spike}) \approx 0.99$.

## Testing for the linear coding model

Within the range $[m_1, m_2]$, we carried out a linear regression, $P(\text{spike}|m) \sim \alpha m + \beta$, for each of the bootstrap samples (see above), and computed the mean and SD of $\alpha$ and $\beta$, as well as the goodness of fit, which was the coefficient of determination, $R^2$ (**Figure 5C,D**).

So far the linear prediction, $m$, in fact described the prediction of the *rate fluctuation around the mean firing rate*. For better illustration in **Figure 5C**, we rescaled and shifted $m$ as $\alpha m + \beta \rightarrow m$ so that $m$ represents the prediction of the actual firing rate.

