## [Decision Letter]

Thank you for submitting your work entitled "Multiplexed coding by cerebellar Purkinje neurons" for consideration by *eLife*. Your article has been favorably evaluated by Timothy Behrens as the Senior editor and three reviewers, one of whom, Fred Rieke, is a member of our Board of Reviewing Editors.

The reviewers have discussed the reviews with one another and the Reviewing Editor has drafted this decision to help you prepare a revised submission.

Consensus review (discussed by all reviewers):

This is an interesting paper that suggests that cerebellar Purkinje cells exhibit multiplexed coding – in which spikes at the onset or offset of pauses in spike activity signify the onset of an eye movement, while spikes part of more regular firing patterns represent eye movement properties. The reviewers felt the paper had the potential to provide key insights into Purkinje cell coding and to resolve some issues in the literature. Several issues, however, limited enthusiasm and need to be dealt with before considering the paper further. These are listed below in rough order of importance.

1) Importance of complex spikes. Complex spikes are described in the Discussion as having effectively no impact on the conclusions of the paper. However, it is not clear to what extent the results presented reflect complex-spike-triggered pauses in Purkinje cell firing (see Reviewer #2 comments). This issue needs to be clarified, presumably by additional analyses.

2) Event selection. All of the reviewers had concerns about how spikes were selected for analysis. The core issue is whether the selection process used in the paper biased the results by focusing on a small subset of spikes and omitting others. Controls to test for such biases and to assess the generality of the reported result are needed. These include a more straightforward analysis to identify pauses (see Reviewer #3 comments) and analysis that are more inclusive of all the spikes (see Reviewer #1 comments).

3) LFP phase reliability and Purkinje cell synchrony. How confident can we be that the reliability of the LFP phase is due to synchrony? The reviewers did not feel this point necessarily needed to be resolved by data from cell populations, but instead that any caveats of the interpretation needed to be discussed carefully.

4) Writing. The results could be presented in a more accessible manner, which will be particularly important give *eLife*'s broad readership. See reviewer comments for specific suggestions.

These, and other, issues are detailed below in the comments from the individual reviewers.

Reviewer #1:

I am not an expert in Purkinje cell coding, so will leave comments about significance of the findings and relation with other work on the cerebellum to the more expert reviewers. The comments below deal specifically with more general coding issues and overall presentation of the work. My main concerns had to do with the selection of data to analyze, and how that selection influences the results and interpretation of the findings as well as some writing issues.

Data selection.

I was concerned about several issues about the data included or excluded from analysis. First, in a subset of recordings (subsection “Cerebellar LFP and single PC firing correlate to saccadic eye movements”, second paragraph) spike times and eye velocity exhibited significant correlations. The other recordings were apparently omitted from subsequent analysis. Is that because these are likely cells that do not participate in the control of eye movements? It would be helpful to indicate why that selection of data is justified.

Second, related to above, some selection of recordings was apparently made based on LFP coherence (subsection “Pause spikes and their correlated LFP component encode the timing of eye motion”, second paragraph) – but the details are also unclear.

Third, the identification of pause spikes and regularly firing spikes omits the majority of the recorded spikes. Given the large differences in properties of the pause and regular spikes, the question of what the other spikes are doing comes up immediately. I think this is a key issue with respect to the multiplexing concept – is there a continuum between the extremes represented by the pause and regular spikes, or is there a more discrete transition? I think of multiplexing as being about a set of discrete signals rather than a continuum. It is important to know how much the results extend to encompass all of the Purkinje cell spikes. This issue comes up in particular with regard to Figure 5.

Writing.

I think the writing can be improved in several places. A partial list follows:

Abstract, second sentence could be split into two.

Abstract, last sentence unclear.

In the third paragraph of the subsection “Pause spikes couple to the LFP strongly and specifically to the β/γ band”, last sentence "This effect […]" is unclear.

In the last paragraph of the subsection “Pause spikes couple to the LFP strongly and specifically to the β/γ band”, define pairwise phase consistency in a sentence.

In the last paragraph of the subsection “Pause spikes and their correlated LFP component encode the timing of eye motion”, explain estimate of extra coincident events in a bit more detail.

In the last sentence of the subsection “Pause spikes and their correlated LFP component encode the timing of eye motion” – I'm not sure why you say that the fidelity depends on the local network – perhaps "reflects" is a better word? What if a set of nearby cells received highly synchronized inhibitory input (not necessarily local) and hence generated high fidelity, synchronized pauses?

In the third paragraph of the subsection “PC firing rate linearly encodes eye motion, mostly by regular spikes”, you describe the pause spikes as exhibiting a high degree of nonlinearity, but it seems to me like that the firing rate is just not very dependent on eye movements. In other words, how do you separate nonlinearity from lack of correlation?

The third paragraph of the Discussion seems a bit repetitive.

In the sixth paragraph of the Discussion, "Exclusion of the complex spikes causes minuscule.…" – can you be more specific about what minuscule means?

Reviewer #2:

This manuscript tackles the very general question of how neurons encode information in their spike trains, and, in particular, the contribution of coding by the precise time of spikes within a train vs. a rate code. The experimental systems used by the authors (cerebellum and saccadic eye movements) offer some powerful advantages for analyzing this issue. The manuscript builds on the De Schutter group's previous work suggesting that brief, synchronized pauses in Purkinje cell firing may play a key role in cerebellar function. The current manuscript expands significantly on what had been done previously by 1) building a tighter link to behavior by examining pauses in the context of saccadic eye movement behavior and 2) examining the relationship between the LFP and the spikes bounding the pause.

The main conclusion is that the pause-related spikes are synchronized and encode movement onset, whereas the other, "regular" spikes encode movement kinematics.

I think this paper makes a significant contribution to the field. The idea that different spikes within a train may encode different information and be read out differently (through synchronized action with other cells vs. rate averaging) should be of broad interest to the neuroscience community.

There are three main areas where I have questions or concerns.

First, the authors claim in the Discussion "Exclusion of the complex spikes caused minuscule changes in our results." This is an important issue that should be more thoroughly documented to rule out that the pause spikes are not reflecting the effect of complex spikes. Pause beginning spikes are defined so that they are 10% of all spikes. Complex spikes are nearly 2% of all spikes, so could potentially account for ~20% of the pause beginning spikes-if pause beginning spikes associated with a complex spike are removed, do the results still hold? If so, that would strengthen the authors' claim that the effects they report are not related to the complex spikes. But the opposite result would also be interesting.

Second, it would be nice to see a more systematic analysis of different frequency bands in the LFP to determine whether it is really just the 15-42 Hz band that couple with the pause spikes.

Based on Figure 5, the authors conclude that the regular spikes more linearly encode eye velocity than the pause spikes. Although this appears to be true, it also appears that there is a nonlinearity in the encoding of eye velocity by the regular spikes, which is compensated by the pause spikes in a way that makes the population as a whole more linear re eye velocity. Is this a reasonable interpretation? If so, it seems worth mentioning.

Reviewer #3:

The manuscript by Hong et al. describes analyses of Purkinje neuron spike trains that shed light on the potential population encoding strategies of this cell population with respect to movement kinematics. They analyze LFPs and simple spikes during saccadic eye movements and find that LFP measurements are more tightly correlated with eye movement onset than single-spike trains, and that long interspike interval – "pauses" – phase-locked with these predictive LFPs. The conclusion from these pause-related data is that pauses encode saccade onsets. On the other hand, overall firing rate is highly correlated with eye velocity, with the "regular" spikes within this population. Pairing these observations, the authors conclude that multiplexed coding occurs with Purkinje firing.

I like the direction taken here and find most of the analyses important. There are some important missing pieces of information, however, that make evaluating the findings challenging and therefore a strong recommendation for accepting or rejecting the manuscript is premature. Of paramount importance is the selection of spikes beginning long pauses: "we selected 15% of largest AI values. […] then removed 25% of the spikes with the shortest pause ISI." Since this process operates on ISI ratios, it is unclear what the ISI durations are that are being analyzed as pauses. Moreover, because this analysis relies on ratios and not overall ISI duration, (which would seem a more natural criterion for long ISIs), it is selecting for essentially bursty sequences of short-then-longish spikes. Interpreting the data through this light seems different than interpreting it through the lens of 'pauses' or longest ISIs. While I acknowledge that there may be no perfect solution to identifying pauses, I am not provided enough information to evaluate their claims because I don't know if the pauses they have chosen are unique in any way, or are alternatively unique in ways they do not highlight. In addition to the distribution of 'pause' ISIs vs. all ISIs, a plot of ISI_n_ vs. ISI_n+1_ would provide non-ratioed data that may help reveal unique characteristics of the 'pause' population.

An important assumption to the interpretation of the data advanced by the study is that the LFP correlation with pause onset spikes reflects synchronous firing. Since LFPs reflect synaptic and spiking currents, it may be that correlation detected is simply a readout of the fact that pushing the ISI off of the basal firing rates reflects synaptic input, which is also reflected in the LFP. I am not confident that any data analysis short of multi single-unit array recordings could resolve whether single spikes are temporally aligned at the onset of a pause to the degree necessary for synchrony coding in the nuclei, but I could be persuaded by experts of LFP interpretation.

Overall I think this manuscript could really help resolve an ongoing debate and find many of their observations tantalizing. The information and caveats I raise above prevent me from providing overwhelming support.

---

## [Author Response]

*This is an interesting paper that suggests that cerebellar Purkinje cells exhibit multiplexed coding – in which spikes at the onset or offset of pauses in spike activity signify the onset of an eye movement, while spikes part of more regular firing patterns represent eye movement properties. The reviewers felt the paper had the potential to provide key insights into Purkinje cell coding and to resolve some issues in the literature. Several issues, however, limited enthusiasm and need to be dealt with before considering the paper further. These are listed below in rough order of importance.*

1) Importance of complex spikes. Complex spikes are described in the Discussion as having effectively no impact on the conclusions of the paper. However, it is not clear to what extent the results presented reflect complex-spike-triggered pauses in Purkinje cell firing (see Reviewer #2 comments). This issue needs to be clarified, presumably by additional analyses.

In the revised manuscript, we have included expanded statistics and discussion about the impact of complex spikes on our results (see also our response to Reviewer #2). We found that the effect of complex spikes was almost negligible, mainly because pauses triggered by complex spikes have different characteristics from simple-spike pauses that we selected.

This is demonstrated in Figure 6 based on the cell used in Figure 2. In A(included as Figure 2—figure supplement 1 in the revision), we plot the asymmetry index (AI) versus post-spike ISI where “pauses” selected by our criterion are in light blue, complex spike pauses in orange, and the rest in purple. First, the AI distribution of complex spikes was slightly shifted to the right but widely spread. In all cells, complex spikes had a larger average AI (complex: 0.36 ± 0.12, simple: 5.9 × 10^-4^ ± 7.7 × 10^-4^, mean ± SD) whereas their standard deviations were comparable (complex: 0.36 ± 0.05, simple: 0.31 ± 0.04). Therefore, many complex spikes had smaller AI than the thresholds. Second, durations of complex-spike pauses did not increase as rapidly as those of simple spike pauses with the asymmetry index. In all the cells, the correlation between AI and log_10_(ISI) was significantly larger for simple spikes than complex spikes (complex: 0.24 ± 0.15, simple: 0.66 ± 0.03, p = 2.84 × 10^-12^, Wilcoxon rank-sum test), which made the complex-spikes pauses poorly predicted by the asymmetry index. This led to only a small overlap between the orange and light blue dots. In all the cells, the overlap, or the fraction of the complex spike-triggered pauses in selected pauses, was only 4.3 ± 2.5% (B).

Author response image 1.**DOI:**
http://dx.doi.org/10.7554/eLife.13810.016

Even in the few cases where complex spikes contributed more than 10% of selected pauses, removing complex spikes improved our signal-to-noise ratio. This is because the LFP triggered by complex spikes have a large positive peak while other pause-initiating simple spikes tend to be associated with a sharp negative peak, as shown in the Figure 7 (PauseI, PauseT, and Regular stand for pause-initiating, -terminating, and regular spikes, respectively).

Author response image 2.**DOI:**
http://dx.doi.org/10.7554/eLife.13810.017

In the revised version, we write in Results “PCs fire regularly most of the time, but occasionally pause”:

“Although complex spikes triggered pauses, their contribution to pauses in our study remained limited. […] For this reason, we did not include complex spikes and their associated pauses in our analysis beyond this point.”

Also in Discussion:

“We also found that the effect of complex spikes was minimal since pauses triggered by complex spikes (Latham and Paul 1970) had a distinct distribution compared to simple-spike pauses (Figure 2—figure supplement 1). The overall impact of complex spikes was negligible, most probably because none of our tasks involved sensorimotor learning where complex spikes are crucial (Catz et al. 2005; Medina and Lisberger 2008), as they tend to occur after saccades, triggered by significant saccade errors (Herzfeld et al. 2015).”

2) Event selection. All of the reviewers had concerns about how spikes were selected for analysis. The core issue is whether the selection process used in the paper biased the results by focusing on a small subset of spikes and omitting others. Controls to test for such biases and to assess the generality of the reported result are needed. These include a more straightforward analysis to identify pauses (see Reviewer #3 comments) and analysis that are more inclusive of all the spikes (see Reviewer #1 comments).

We checked whether including more spikes would change our results. In the selection criterion for pause spikes described in the original submission, we chose 15% of the spikes with larger CV_2_ values and removed 25% of those that were associated with shorter pauses. Previously we used a simpler criterion, the CV_2_ criterion, which gave very similar results as in Figure 8, which shows the STA_LFP_s for the same cell as in Figure 3. Because we received feedback that this resulted in the inclusion of rather short pauses, we added the short pause removal step to the selection criterion.

Author response image 3.**DOI:**
http://dx.doi.org/10.7554/eLife.13810.018

As an additional control, we varied the fraction of the initial choice from 10% to 50% by steps of 10%, and also applied the 25% removal of short pauses. Therefore, the fraction of pause spikes varied from 7.5% to 37.5% in increments of 7.5%. Figure 3—figure supplement 1 shows how the STA_LFP_ in Figure 3 varied with different number of spikes chosen. In pause spikes, the fast fluctuating initial part of the STA_LFP_ is quite invariant in shape while the amplitude slightly decreases as we include more spikes. In the regular spike case, there was almost no difference.

For all data, we computed STAs specifically with the β-γ LFP (Figure 3—figure supplement 1) and their amplitude relative to the STA of all spikes (as in Figure 3). Figure 3—figure supplement 1 shows how they change when more spikes are included. Amplitudes for the pause spike case again tended to decrease with more spikes, but they were still much larger than those for regular spikes. Therefore, conservatively speaking, if we choose about 25% of all spikes as pause-initiating or -terminating spikes, based on our criterion, their behavior will be qualitatively similar to the more extreme selections of ~10% in the paper. However, the distinction shows a gradient rather than a clear cut.

A similar analysis was already included in the original submission for linear encoding of motion kinematics by regular spikes (Figure 5—figure supplement 1).

We would like to stress that the lack of a clear separation between pause and regular spikes in our data – which is compatible with the multiplexing hypothesis – was one of the motivations to compare the three extreme cases in our analysis. This also addresses Reviewer #3’s comments about the pause criterion: we have tried multiple criteria for pauses, and as long as we can obtain a sufficient sample size in each group, all of them resulted in qualitatively more or less the same results. Therefore, we decided to settle on one of the simplest ones.

Following the request of Reviewer #3, we made a pre-spike ISI versus post-spike ISI plot (same cell as Figure 3) (A) (included as Figure 2—figure supplement 1). Here, the red, cyan, and black dots represent, pause-initiating, -terminating, and regular spikes, based on our criterion, while the gray dots are the rest of the spikes. Contours represent log_10_(density) and become quickly triangular as we move to the tail part of the ISI distribution (pauses), which is captured by the selected pause spikes. We previously wrote “The minimal pause duration varied with the average firing rate of the cell, but was in general ~20% larger than the ISI corresponding to the mean firing rate.” We see this more explicitly in the ISI histograms, normalized by their peaks (B). Post-spike ISIs of pause-initiating spikes (red solid line) are significantly larger than those corresponding to the mean firing rate (18.4 ms, blue dot and vertical line). Pre-spike ISIs (red dotted line) are the shortest of all, but it is still within the overall spike distribution that peaks around 10-15 ms.

Author response image 4.**DOI:**
http://dx.doi.org/10.7554/eLife.13810.019

3) LFP phase reliability and Purkinje cell synchrony. How confident can we be that the reliability of the LFP phase is due to synchrony? The reviewers did not feel this point necessarily needed to be resolved by data from cell populations, but instead that any caveats of the interpretation needed to be discussed carefully.

Unfortunately, we don’t have any data to resolve how much of the LFP signal originates from Purkinje cell synchrony and we do not wish to suggest that Purkinje cell synchrony is a direct cause of the LFP signal. The contribution of other cells, providing inputs to the Purkinje neurons, could also be substantial. Therefore, we instead argue that Purkinje cell pausing reflects this temporary, sharp, circuit-wide activity and that accordingly, synchrony of pause spikes may be a part of it. Previous studies observed similar LFP phenomena in the granular layer (Morisette and Bower, Exp Brain Res, 1996; Roggeri et al., J Neurosci 2008), which seems to arise from dense synchronous activation of granule cells (Diwakar et al., PLOS One 2011). We do not have any means to probe whether our LFP signal was conducted from the granular layer, but even if not, we think it is likely that such a synchronous activation of mossy fibers and granule cells could spread to many cells, including molecular layer interneurons and ultimately Purkinje cells. Strong activation of interneurons will provide feedforward inhibition to Purkinje cells (Mittmann et al., J Physiol, 2004). This inhibition can cause simple-spike pausing (Mittmann and Häusser, J Neurosci, 2007) while significantly contributing to the LFP signal.

In the revised manuscript, we have now modified the Discussion to state:

“We did not attempt to resolve the origin of the time encoding LFP signal due to limitations of the experimental setup. […] If simultaneous activation of local MLIs (and/or their synaptic inputs to the local population of PCs) contributes to the LFP, this would explain our observed correlation of the fast LFP signal and pauses in PCs.”

4) Writing. The results could be presented in a more accessible manner, which will be particularly important give eLife's broad readership. See reviewer comments for specific suggestions.

We made all the changes that reviewers suggested. Also, to improve readability of our manuscript extensively, OIST’s technical editor has twice reviewed the manuscript. We have substituted the phrases “pause-initiating” and “pause-terminating” for “pause beginning” and “pause ending.” Because the editor streamlined and simplified the language, we believe that our paper is now as accessible as possible, given the subject’s technical complexity.

*These, and other, issues are detailed below in the comments from the individual reviewers.*

*Reviewer #1:*

*I am not an expert in Purkinje cell coding, so will leave comments about significance of the findings and relation with other work on the cerebellum to the more expert reviewers. The comments below deal specifically with more general coding issues and overall presentation of the work. My main concerns had to do with the selection of data to analyze, and how that selection influences the results and interpretation of the findings as well as some writing issues.*

*Data selection.*

*I was concerned about several issues about the data included or excluded from analysis. First, in a subset of recordings (subsection “Cerebellar LFP and single PC firing correlate to saccadic eye movements”, second paragraph) spike times and eye velocity exhibited significant correlations. The other recordings were apparently omitted from subsequent analysis. Is that because these are likely cells that do not participate in the control of eye movements? It would be helpful to indicate why that selection of data is justified.*

The reviewer is right about our assumption that cells with non-significant responses do not participate in encoding eye motion. The eye saccade-related area in the cerebellar vermisVIc (oculomotor vermis, OMV) is limited to a tiny subregion of lobuli VIc and VIIA, the OMV proper (Prsa and Thier, Eur J Neurosci, 2011), and the classical studies on which our study builds used activity-based techniques to identify eye motion-related responses, such as 1) identifying saccade-related background activity, 2) saccade-related Purkinje cell discharge, and 3) saccade-triggering microstimulation, to locate this subregion (Ohtsuka and Noda, Neurosci Res, 1992). Our criteria based on significant correlations of eye speed to LFP and Purkinje cell firing are equivalent to 1 and 2.

We explain this in the updated manuscript:

“We identified eye saccade-related neurons based on CCF_LFP-EV_ and CCF_Spike-EV_, because this is essentially identical to the classical method used to localize eye motion-sensitive cells (Ohtsuka and Noda 1992). Differences in how the LFP and PC spikes relate…”

*Second, related to above, some selection of recordings was apparently made based on LFP coherence (subsection “Pause spikes and their correlated LFP component encode the timing of eye motion”, second paragraph) – but the details are also unclear.*

We regret that this part was not clear. We did not sub-select data sets based on the LFP coherence, but showed results for recordings from all 34 cells. Therefore, for example, there is a wide range of variation in LFP phase reliability (=LFP coherence across trials), while a majority of data shows statistical significance compared to the controls (Figure 4).

Third, the identification of pause spikes and regularly firing spikes omits the majority of the recorded spikes. Given the large differences in properties of the pause and regular spikes, the question of what the other spikes are doing comes up immediately. I think this is a key issue with respect to the multiplexing concept – is there a continuum between the extremes represented by the pause and regular spikes, or is there a more discrete transition? I think of multiplexing as being about a set of discrete signals rather than a continuum. It is important to know how much the results extend to encompass all of the Purkinje cell spikes. This issue comes up in particular with regard to Figure 5.

Our original intent in selecting small subsets of spikes was that we wanted to probe differences between extreme cases. We performed extra analyses by including more spikes in the selection, which is described in our response to consensus review 2. To summarize our conclusion, our observations are quite robust to variation. The transition seems quite continuous, just like many other aspects of our data. For example, there is no discrete transition between the regular ISI and longer pauses, but they form a unimodal distribution with a long tail (Figure 2 in the manuscript). The reliability of pause spikes in encoding saccade onset is not distinguishably higher than that of regular spikes in some PCs, but it varies with LFP reliability (Figure 4). We believe that for efficient multiplexing it may be beneficial if a population of PCs differ in the degree to which each behavior is expressed. This should be more advantageous in encoding eye motion since there is a continuous gradient from short and tiny to long and large saccadic eye motion. For effective coding, Purkinje cell pauses should have variable sizes and effects.

Writing.

*I think the writing can be improved in several places. A partial list follows:*

*Abstract, second sentence could be split into two.*

*Abstract, last sentence unclear.*

*In the third paragraph of the subsection “Pause spikes couple to the LFP strongly and specifically to the β/γ band”, last sentence "This effect.…" is unclear.*

*In the last paragraph of the subsection “Pause spikes couple to the LFP strongly and specifically to the β/γ band”, define pairwise phase consistency in a sentence.*

In the last paragraph of the subsection “Pause spikes and their correlated LFP component encode the timing of eye motion”, explain estimate of extra coincident events in a bit more detail.

We thank reviewer for these specific suggestions. We have revised the manuscript accordingly.

In the last sentence of the subsection “Pause spikes and their correlated LFP component encode the timing of eye motion” – I'm not sure why you say that the fidelity depends on the local network – perhaps "reflects" is a better word? What if a set of nearby cells received highly synchronized inhibitory input (not necessarily local) and hence generated high fidelity, synchronized pauses?

We agree with the reviewer that “reflects” is better. Highly synchronized inhibitory inputs are definitely possible, and can be a source of the LFP signal related to pauses.

In the third paragraph of the subsection “PC firing rate linearly encodes eye motion, mostly by regular spikes”, you describe the pause spikes as exhibiting a high degree of nonlinearity, but it seems to me like that the firing rate is just not very dependent on eye movements. In other words, how do you separate nonlinearity from lack of correlation?

We used “nonlinearity” to simply describe the lack of a linear relationship between the linear prediction and the actual firing rate. The reason why the firing rate of pause spikes does not look very dependent on eye movements is that we computed motion features (Figure 5) from all spikes. Pause spikes do correlate with eye motion (e.g. Figure 4), but very differently from the full spike train for which the correlation to eye motion is dominated by regular spikes, as we argued in Figure 5—figure supplement 1 and the related text. Therefore, the firing rate of pause spikes has a weak and nonlinear dependence along the direction of those motion features. To further illustrate this, we plotted a figure (included as an inset of Figure 5—figure supplement 1 in the revision) how similar (measured by the correlation ρ) the CCF_Spike-EV_ for pause spikes is to that of the full spike train, as we change thresholds for pause spikes, similarly to Figure 5—figure supplement 1. In contrast to Figure 5—figure supplement 1, the CCF_Spike-EV_ of pause spikes is still unmatched to that of all spikes, even when we classify more than 50% of the spikes as pause spikes.

In the revised text, we clarified this point in Results – PC firing rate linearly encodes eye motion, mostly by regular spikes:

“On the other hand, pause spikes (both pause-initiating and -terminating spikes) showed high degrees of nonlinearity (*R*^2^ = 0.39 ± 0.25) as their firing rate did not vary as steeply as for regular spikes (Figure 5). This suggests that information encoded by the whole firing rate is very different from eye movement features related to pause spikes such as onsets (Figure 4), but this information can be captured well by a subset of the full spike train, regular spikes.”

and in next paragraph:

“Note that *ρ* for pause spikes behaved very differently; it was not significant for reasonable fractions of pause spikes (Figure 5—figure supplement 1 inset).”

The third paragraph of the Discussion seems a bit repetitive.

Fixed.

In the sixth paragraph of the Discussion, "Exclusion of the complex spikes causes minuscule.…" – can you be more specific about what minuscule means?

This part has been replaced with an expanded discussion about complex spikes (see our response to consensus review 1).

*Reviewer #2:*

*This manuscript tackles the very general question of how neurons encode information in their spike trains, and, in particular, the contribution of coding by the precise time of spikes within a train vs. a rate code. The experimental systems used by the authors (cerebellum and saccadic eye movements) offer some powerful advantages for analyzing this issue. The manuscript builds on the De Schutter group's previous work suggesting that brief, synchronized pauses in Purkinje cell firing may play a key role in cerebellar function. The current manuscript expands significantly on what had been done previously by 1) building a tighter link to behavior by examining pauses in the context of saccadic eye movement behavior and 2) examining the relationship between the LFP and the spikes bounding the pause.*

*The main conclusion is that the pause-related spikes are synchronized and encode movement onset, whereas the other, "regular" spikes encode movement kinematics.*

*I think this paper makes a significant contribution to the field. The idea that different spikes within a train may encode different information and be read out differently (through synchronized action with other cells vs. rate averaging) should be of broad interest to the neuroscience community.*

*There are three main areas where I have questions or concerns.*

First, the authors claim in the Discussion that "Exclusion of the complex spikes caused minuscule changes in our results." This is an important issue that should be more thoroughly documented to rule out that the pause spikes are not reflecting the effect of complex spikes. Pause beginning spikes are defined so that they are 10% of all spikes. Complex spikes are nearly 2% of all spikes, so could potentially account for ~20% of the pause beginning spikes-if pause beginning spikes associated with a complex spike are removed, do the results still hold? If so, that would strengthen the authors' claim that the effects they report are not related to the complex spikes. But the opposite result would also be interesting.

We thank the reviewer for calling our attention to this important issue. We have added a more detailed account of why the effect of complex spikes is limited in our response to consensus review 1.

Second, it would be nice to see a more systematic analysis of different frequency bands in the LFP to determine whether it is really just the 15-42 Hz band that couple with the pause spikes.

Since the STA_LFP_ of pause spikes fluctuates transiently, it triggers a high coherence in a wide frequency band, following the time-frequency uncertainty principle. However, the coherence peak was located at higher frequencies (28.2 ± 5.7 Hz, 29.2 ± 4.6 Hz for pause-initiating and -terminating spikes, respectively), which was the most consistent pattern that we observed. In the lower frequency band (< 15Hz), coherence peaks tend to be smaller, by a factor of 0.36 ± 0.66 and 0.16 ± 0.23 for pause-initiating and -terminating spikes, respectively. This made the average coherence less than 0.05 in the lower frequency band, which predicted much weaker phase-locking in this band.

Author response image 5.**DOI:**
http://dx.doi.org/10.7554/eLife.13810.020

Our measure of spike-LFP phase locking, pairwise phase consistency (PPC; Figure 3), indeed showed this. Figure 3—figure supplement 3 shows the PPC of each frequency band and type of spikes, normalized by the β-γ case. The PPC sharply dropped as we moved to lower frequency bands such as the low β (12-15 Hz), theta (4-10 Hz), and δ (0-4 Hz), except for a few outliers, in pause-spike cases. Compared to this, regular spikes showed an inconsistent pattern.

In summary, the 15-42 Hz band was the only frequency band in which we observed robust coupling to pause spikes.

Based on Figure 5, the authors conclude that the regular spikes more linearly encode eye velocity than the pause spikes. Although this appears to be true, it also appears that there is a nonlinearity in the encoding of eye velocity by the regular spikes, which is compensated by the pause spikes in a way that makes the population as a whole more linear re eye velocity. Is this a reasonable interpretation? If so, it seems worth mentioning.

It is indeed reasonable since we are testing how the LN model constructed from all spikes can predict the firing rate of each group. Therefore, if the coding by all spikes is really linear, nonlinearity in one group should be compensated by those of other groups. As the reviewer remarks, we show that linearity is indeed slightly smaller for regular spikes, and pause spikes should compensate. The revised text now reads,

“Nevertheless, pause spikes can make small contributions to rate coding. […] Pause spikes, which by definition represent fast rate changes, should compensate for this. However, overall rate coding is dominated by regular spikes.”

*Reviewer #3:*

*The manuscript by Hong et al. describes analyses of Purkinje neuron spike trains that shed light on the potential population encoding strategies of this cell population with respect to movement kinematics. They analyze LFPs and simple spikes during saccadic eye movements and find that LFP measurements are more tightly correlated with eye movement onset than single-spike trains, and that long interspike interval – "pauses" – phase-locked with these predictive LFPs. The conclusion from these pause-related data is that pauses encode saccade onsets. On the other hand, overall firing rate is highly correlated with eye velocity, with the "regular" spikes within this population. Pairing these observations, the authors conclude that multiplexed coding occurs with Purkinje firing.*

*I like the direction taken here and find most of the analyses important. There are some important missing pieces of information, however, that make evaluating the findings challenging and therefore a strong recommendation for accepting or rejecting the manuscript is premature. Of paramount importance is the selection of spikes beginning long pauses: "we selected 15% of largest AI values.[…] then removed 25% of the spikes with the shortest pause ISI." Since this process operates on ISI ratios, it is unclear what the ISI durations are that are being analyzed as pauses. Moreover, because this analysis relies on ratios and not overall ISI duration, (which would seem a more natural criterion for long ISIs), it is selecting for essentially bursty sequences of short-then-longish spikes. Interpreting the data through this light seems different than interpreting it through the lens of 'pauses' or longest ISIs. While I acknowledge that there may be no perfect solution to identifying pauses, I am not provided enough information to evaluate their claims because I don't know if the pauses they have chosen are unique in any way, or are alternatively unique in ways they do not highlight. In addition to the distribution of 'pause' ISIs vs. all ISIs, a plot of ISI_n_ vs. ISI_n+1_ would provide non-ratioed data that may help reveal unique characteristics of the 'pause' population.*

For the plots, please see our response to consensus review 2. The ISI_n_ versus ISI_n+1_ plot shows that the distribution of (ISI_n_, ISI_n+1_) is stretched along the x- and y-axes, particularly for longer ISIs. In combination with the ISI distribution with pause ISIs vs. all ISIs, this shows that indeed our criterion selects the tail of the distribution. Selected pauses are indeed longer than average ISIs, but they are not all at extreme durations.

Choosing the longer ISIs following shorter ones, by using ratios, is important to us, since they occurred at saccade onsets in many Purkinje cells. Some Purkinje cells show relatively long periods of very low firing far after saccade onset, and long ISIs there should have a different role than our pauses in encoding eye movements, such as contributing to a rate code. Long pauses that follow short ISI are also likely to release cerebellar nucleus neurons from inhibition, which has been shown to encode motion onsets (e.g., Lee et al., Neuron, 2015).

*An important assumption to the interpretation of the data advanced by the study is that the LFP correlation with pause onset spikes reflects synchronous firing. Since LFPs reflect synaptic and spiking currents, it may be that correlation detected is simply a readout of the fact that pushing the ISI off of the basal firing rates reflects synaptic input, which is also reflected in the LFP. I am not confident that any data analysis short of multi single-unit array recordings could resolve whether single spikes are temporally aligned at the onset of a pause to the degree necessary for synchrony coding in the nuclei, but I could be persuaded by experts of LFP interpretation.*

We think it is unlikely that the LFP reflects only synaptic events in recorded cells. For example, Figure 1 and Figure 1—figure supplement 1 show that, although the LFP signal more or less stays the same for different saccade angles, the direction of firing modulation can flip completely. Our conclusion is that the LFP signal should reflect a certain network event that involves many neurons (but not necessarily only Purkinje neurons). We provide a more detailed discussion in our response to consensus review 3.